# Vertebral Body Tethering in AIS Management—A Preliminary Report

**DOI:** 10.3390/children10020192

**Published:** 2023-01-20

**Authors:** Aurélien Courvoisier, Alice Baroncini, Clément Jeandel, Clémentine Barra, Yan Lefevre, Federico Solla, Richard Gouron, Jean-Damien Métaizeau, Marie-Christine Maximin, Vincent Cunin

**Affiliations:** 1TIMC, University Grenoble Alpes, CNRS, UMR 5525, VetAgro Sup, Grenoble INP, CHU Grenoble Alpes, 38000 Grenoble, France; 2Grenoble Alps Scoliosis and Spine Center, Grenoble Alps University Hospital, Bvd de la Chantourne, CEDEX 09, 38043 Grenoble, France; 3Department of Orthopaedics, RWTH Uniklinik Aachen, Pauwelsstraße 30, 52074 Aachen, Germany; 4Service d’Orthopédie Pédiatrique, Hôpital Femme Mère Enfant, Hôpitaux civils de Lyon, 59 Bd Pinel, 69500 Bron, France; 5Service d’Orthopédie Pédiatrique, CHU de Bordeaux, Pl. Amélie Raba Léon, 33000 Bordeaux, France; 6Service d’Orthopédie Pédiatrique, Fondation Lenval, 57 Avenue de la Californie, 06200 Nice, France; 7Service d’Orthopédie Pédiatrique, CHU d’Amiens, Université Picardie Jules Verne, Chemin du Thil, CS 52501, CEDEX 1, 80025 Amiens, France; 8Service d’Orthopédie Pédiatrique, Centre Hospitalier Universitaire F.Mitterrand Dijon-Bourgogne, 14 Rue Paul Gaffarel, 21000 Dijon, France

**Keywords:** vertebral body tethering, growth modulation, idiopathic scoliosis

## Abstract

Vertebral Body Tethering (VBT) is a recently developed surgical technique for the treatment of progressive and severe scoliosis in patients with significant growth potential. It has been used since the first exploratory series, which showed encouraging results on the progressive correction of the major curves. This study reports on a retrospective series of 85 patients extracted from a French cohort, with a follow-up at a minimum of two years after a VBT with recent screws-and-tether constructs. The major and compensatory curves were measured pre-operatively, at the 1st standing X-ray, at 1 year, and at the last available follow-up. The complications were also analyzed. A significant improvement was observed in the curve magnitude after surgery. Thanks to growth modulation, both the main and the secondary curves continued to progress over time. Both the thoracic kyphosis and lumbar lordosis remained stable over time. Overcorrection occurred in 11% of the cases. Tether breakage was observed in 2% of the cases and pulmonary complications in 3% of the cases. VBT is an effective technique for the management of adolescent idiopathic scoliosis patients with residual growth potential. VBT opens an era of a more subtle and patient-specific surgical management of AIS that considers parameters such as flexibility and growth.

## 1. Introduction

Vertebral Body Tethering (VBT) is a recently developed surgical technique for the treatment of progressive and severe scoliosis in patients with significant growth potential. The principle of the technique relies on vertebral growth modulation, which is a direct application of the Hueter–Volkman law on the spine and was first described by Crawford in 2010 [1].

Specific medical implants have been developed for VBT and are composed of screws that are placed on the lateral aspect of each vertebral body, on the convex side of the curve, and a single synthetic tether spans between the screws. Tension is then applied to the tether to obtain a peri-operative correction, which is equivalent to a “brace-effect”. The correction improves over time due to a “convex side epiphysiodesis effect” if the patient has significant remaining growth. Flexibility and growth, in this order, are the two main pre-requisites for a successful VBT. VBT is a minimally invasive procedure performed by most surgeons through thoracoscopy or mini-open thoracotomy [2]. The two main objectives of the technique are to avoid fusion and to maintain spine flexibility.

Since the first exploratory series, which showed encouraging results [3,4,5] on the progressive correction of the major curve, the indications have been refined and the longer-term results (2 to 5 years) consolidated the first observations [1,3,4,5,6,7,8,9,10,11,12,13,14,15,16,17,18,19,20,21,22,23]. These retrospective series also showed the maintenance of the mobility of the instrumented spine and a neutral or positive effect on sagittal balance [24], and they confirmed the low risk of immediate complications [25,26,27,28,29]. However, other studies have reported unpredictable results and a higher number of revision surgeries in patients treated with VBT compared to those treated with posterior fusion [26,30]. Overall, the authors agree on the power of growth modulation and studies are being conducted to optimize the eligibility criteria for VBT in an effort to limit the rate of revisions [6,31,32].

Several techniques and implants have been used in our centers in the almost decade-long experience with VBT, with successes and failures. The indications and technical aspects have been rationalized. This study reports on a series of 85 patients with a follow-up at a minimum of two years after a VBT with recent screws-and-tether constructs.

## 2. Materials and Methods

The present bi-centric retrospective study was performed according to the French reference methodology MR-004 between 2017 and 2020 and authorized by our institution. The study has been approved by the SFAR Ethical Committee: IRB 00010254-2021-202. All parents and patients received an information letter. The present study was conducted according to the STROBE statement [33].

### 2.1. Patient Selection

The inclusion criteria were:-Diagnosis of idiopathic scoliosis from 9 to 14 years old;-Severe curve (>40° for thoracic scoliosis/>35° for lumbar scoliosis);-Skeletal immaturity assessed by a Risser index between 0 and 2;-Surgical treatment using the “vertebral growth modulation” technique using screws and synthetic tether.

Patients who lacked a minimum 2-year follow-up were excluded from the study.

### 2.2. Surgical Technique

#### 2.2.1. Right Thoracic Curves

All procedures were performed through the thoracoscopic approach with one-lung ventilation in the left lateral decubitus. A paravertebral bloc is systematically performed [34]. Four to five incisions on the mid-axillary line (15 mm—trocar size) were typically sufficient to perform the entire procedure. A small approach of the vertebral pleura consisted in coagulation of the segmental vessels in the middle of the lateral aspect of the convex side of each vertebral body. The use of a Pediguard (Spineguard—France), a threaded electronic conductivity device, is systematic in our hands to secure the tap trajectory while limiting radiation exposure [35]. The screws are then inserted in the vertebral body. The screws at each end of the construct are slightly tilted (downward at the upper end and upward at the lower end) to improve the strength of the construct. The cord is progressively placed within the screw heads from the cranial to the caudal end. Curve correction is performed with a combination of contraction maneuvers on the counter torques connected to the screws and tension of the tether. A standard suction bottle chest drain is placed in the thorax and is removed at day 1 [36]. Patients walk at day 1 and are usually discharged at day 2 or 3 post-op.

#### 2.2.2. Lumbar Curves

The procedure is very similar to the one described above, except for the screws at L2 and below. The thoracic screws are placed through the thoracoscopic approach with one-lung ventilation in the right lateral decubitus. Three further thoracoscopic incisions are usually sufficient to insert T11, T12, and L1 screws on the left side. A mini-lumbotomy is then performed to approach the sub-diaphragmatic vertebrae. We usually perform a single tether construct. L2, L3, and L4 screws are placed through a trans-psoas approach. A small incision is performed in the diaphragm pillar close to the spine to slide the cord from the thorax down to the lower screws. A special attention to the genito-femoral nerve needs to be observed when placing the L3 screw. Paresthesia of the proximal and antero-medial aspect of the thigh is frequent after this surgery even when the nerve is not damaged. Post-op medication with Gabapentine is systematic in our patients.

#### 2.2.3. Double Curves

Both the lumbar and the thoracic procedures are sequentially performed (lumbar first), usually on the same day. Selective thoracic VBT is possible in some cases for double curves (Figure 1 and Figure 2). In some cases, spontaneous correction of the lumbar curve is not effective. In those cases, the lumbar procedure is postponed. The neutral vertebra located at the junction is instrumented on both sides.

### 2.3. Post-Operative Management

The first full-spine erect radiograph (EOS–EOS-Imaging–France) is performed at day 2, then at 3, 12, 18 and 24 months post-op, and then once a year until skeletal maturity and at least 5 years post-op.

All patients are braced for 6 weeks after surgery to restrain activity. Soft sports (swimming, biking) are authorized after 6 weeks until 3 months post-op. Full return to sport is authorized after 3 months (with contact sports such as rugby, the time is 12 months).

The device removal is not planned systematically.

### 2.4. Outcomes of Interest

Baseline demographic data such as gender, age at surgery, skeletal maturity (Risser grade and evaluation of the triradiate cartilage—TRC) and Lenke curve type were collected.

The major, instrumented and compensatory curves were measured pre-operatively, at the 1st standing X-ray, at 1 year, and at the last available follow-up using the Cobb method. The thoracic kyphosis and lumbar lordosis were also evaluated at the same timepoints. Radiographic data were collected on the whole spine anteroposterior and lateral whole spine EOS scans (EOS-Imaging).

The duration of the hospitalization and the presence and localization of pain were also recorded.

The rate of the surgical-approach-related (vascular or any organ damage) and implant-related complications (tether breakage, screw pullout) was evaluated. The number of surgical revisions was recorded.

### 2.5. Statistical Analysis

The statistical analysis was performed on Excel (Microsoft). Continuous data were expressed as mean and standard deviation, while the categorical variables were expressed as percentages. A two-sided, paired *t*-test was performed to compare the radiographic data from the 1st standing X-ray with those from the last follow-up to assess the growth modulation effect and the variations in the sagittal parameters. A 95% confidence interval was set for all comparisons (*p* = 0.05).

## 3. Results

### 3.1. Patient Selection and Demographic Data

During the observation period, 87 patients meeting the inclusion criteria were treated with VBT at six institutions. Two patients were excluded because they lacked the minimum required follow-up; so, the data from 85 patients were available for this analysis (Figure 3).

The baseline characteristics of the patients in the series are described in Table 1.

The collected radiographic data are summarized in Table 2. Overall, a significant improvement was observed in the curve magnitude after surgery. Thanks to the growth modulation, both the main and the secondary curves continued to progress over time. Both the thoracic kyphosis and the lumbar lordosis remained stable over time.

### 3.2. Radiological Outcome at Last Follow-Up

At the last follow-up, the patients included at Risser 0 TRC closed or above (82%) had all reached Risser 3 or more. Eleven (13%) patients included at Risser 0 TRC open had also reached Risser 3 or more at the last follow-up. Four (5%) patients with severe but flexible curves included at Risser 0 TRC open were still at Risser 2 with ongoing significant growth.

### 3.3. Complications

The type of complications observed in the cohort and the relative treatment are shown in Table 3.

## 4. Discussion

The main finding of the present study is that VBT allows an initial correction of the major curve, which improves over time, from the first post-operative year (Figure 1, Figure 2, Figure 4 and Figure 5). The current cohort is one of the largest in the published literature with a minimum 2-year follow-up. Our findings support most of the reported outcomes in the largest published series, particularly when comparing the major curve measurements at the last follow-up (Table 4). VBT allows a progressive improvement of major curve measurements from 50° to 20° in patients included at around 12 years old. Other series had smaller major curve improvement, which was related to either a higher age at surgery [37] or to a lack of initial pre-operative correction [3,19,21].

If the residual growth is a fundamental element, the flexibility of the curvature is the essential parameter. The initial intraoperative correction (the “brace” effect) is necessary to trigger the modulation phenomenon. The subtlety of the VBT technique lies in the choice of the initial correction, which depends on the amplitude of the curvature and the growth potential. In comparison, fusion is more predictable because it circumvents these parameters.

Sagittal balance in spine surgical treatments has been better understood and evaluated over the past ten years. Our study confirmed the results of a recent work which showed the positive or neutral effect of the VBT technique on the profile [24] (Figure 2 and Figure 6). The mobility, even partial, of the spine allows it to maintain a capacity for compensation. Thus, profile anomalies might be better absorbed than in fusion patients.

Immediate post-operative complications were rare and benign. They were essentially pulmonary. Aseptic pleural effusions have been reported [25], which were rare in our series. The limited approach to the pleura and the use of a simple suction bottle drain [36] might limit the risk of pleural complications. Recent studies have shown that the short-term lung function is not affected by the development of complications [25].

Overcorrection is now a well-known and predictable phenomenon (here 11%), which concerns the youngest patients with open TRC [6]. It often begins at the bottom of the construct between T10 and L1 for thoracic curves. Optimizing surgical timing will help reduce this complication, which may justify cutting the cord in areas undergoing overcorrection (Figure 4). Tether breakages are commonly reported in the literature in lumbar constructs but are not systematically a source of revision surgery [30]. The data observed in the present study confirm this finding.

The failures of the procedure were linked either to the curves that were too rigid and severe or to an “adding-on” phenomenon resulting from poor selection of the instrumented lower end vertebra. Thus, for long thoracic curvatures, we recommend choosing the most neutral vertebra (L2 or even L3) (Figure 5, Figure 6 and Figure 7).

This work is the result of a learning curve of a recent technique. The selection criteria were broad and experience is now refining them. The main limitations of this study were its retrospective, uncontrolled nature and the absence of long-term data. However, 95% of the patients included in the study have reached Risser 3 or more. The construction of prospective randomized studies in spine surgery is complex but now necessary to compare the different treatments.

It is becoming clearer today that the success of the VBT technique lies as much in the patient selection as in the technical realization. Data are accumulating on the relevance of the technique and its effectiveness in a now well-defined population: short Lenke 1 thoracic curvature, greater than 45°, flexible, Risser 0 (Figure 1) The combination of data from the literature with the results of this French series can make it possible to rationalize the indications and thus progressively integrate VBT into the care of our scoliotic patients.

## 5. Conclusions

VBT is an effective technique for the management of adolescent idiopathic scoliosis patients with residual growth potential. In our view, VBT opens an era of a more subtle and patient-specific surgical management of AIS that considers parameters such as flexibility and growth. The global philosophy is to restore not only a normal spine shape but also a normal function. In this line, with our actual knowledge, revisions may be necessary to fine-tune the treatment. For selected patients, fusion no longer represents the only viable option for severe curves. With more experience, surgeons will learn how to use VBT along with fusion or other treatments to restore the most appropriate function for each patient’s spine.

## Figures and Tables

**Figure 1 children-10-00192-f001:**
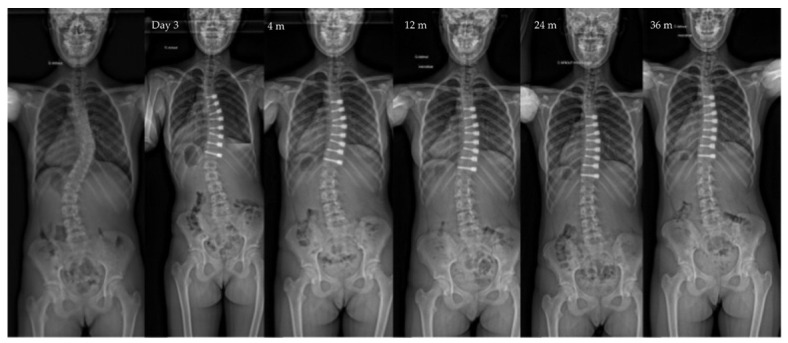
Radiographs of a 12-year-old patient, Risser 0, with a main thoracic curve. The major curve measured 46° pre-operatively, improved to 24° after surgery to 10° at last follow-up (3 y po—Risser 4). Note the spontaneous correction of the lumbar curve over time in this selective VBT.

**Figure 2 children-10-00192-f002:**
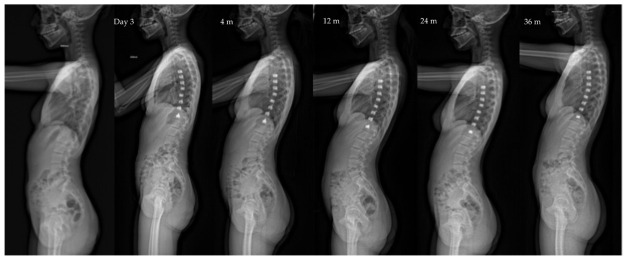
Same patient as Figure 1 Lateral view X-rays throughout follow-up, showing the slight kyphosis improvement and lordosis stability: kyphosis and lordosis are, respectively, 26/55 pre-op, 30/55 at 4 months po, and 35/58 at 36 months po.

**Figure 3 children-10-00192-f003:**
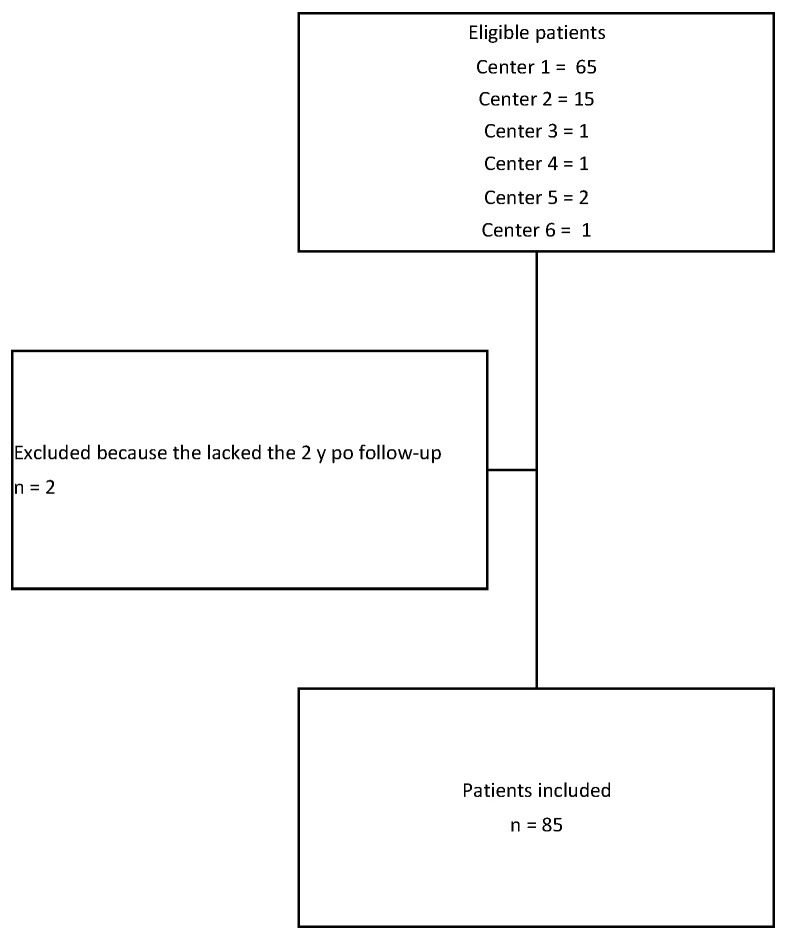
Flowchart of the study.

**Figure 4 children-10-00192-f004:**
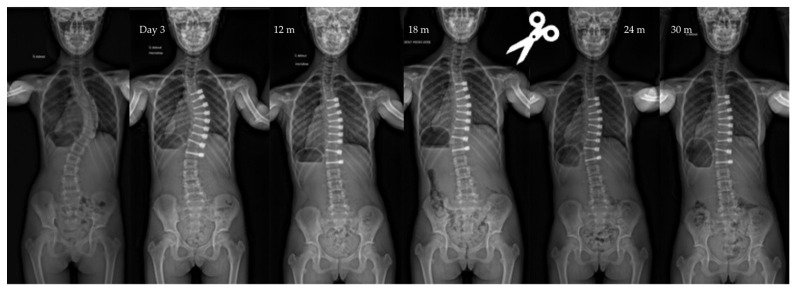
Radiographs of a 11-year-old patient, Risser 0, TRC open, with a main thoracic curve. The curve measured 57° pre-operatively, improved to 26° after surgery, to 11° at last follow-up (2.5 y po). The patient was revised 1.5 po for tether release between T10 and L11. Note the control of the overcorrection after the revision.

**Figure 5 children-10-00192-f005:**
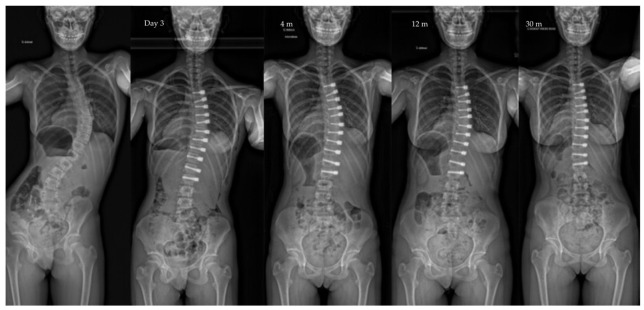
Radiographs of a 12.5-year-old patient, Risser 0, with a main long thoracic curve (lower end vertebra at L3. The major curve measured 57° pre-operatively and improved to 27° after surgery. Note the progressive frontal rebalance due to the thoraco-lumbar growth modulation. At last follow-up (2.5 y po), the major curve was measured at 9°.

**Figure 6 children-10-00192-f006:**
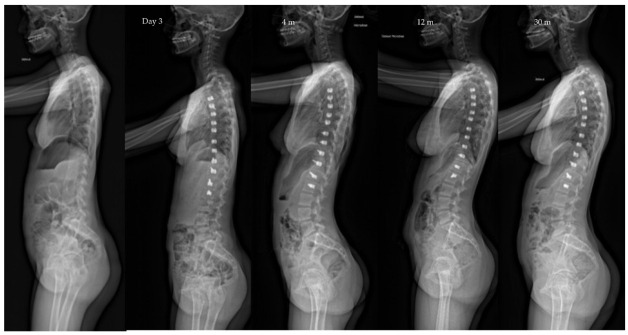
Same patient as Figure 6. Lateral radiographs evolution over the same period of 2.5 years. Note the progressive restoration of the sagittal balance (kyphosis 30° to 40° and lordosis 25° to 38°).

**Figure 7 children-10-00192-f007:**
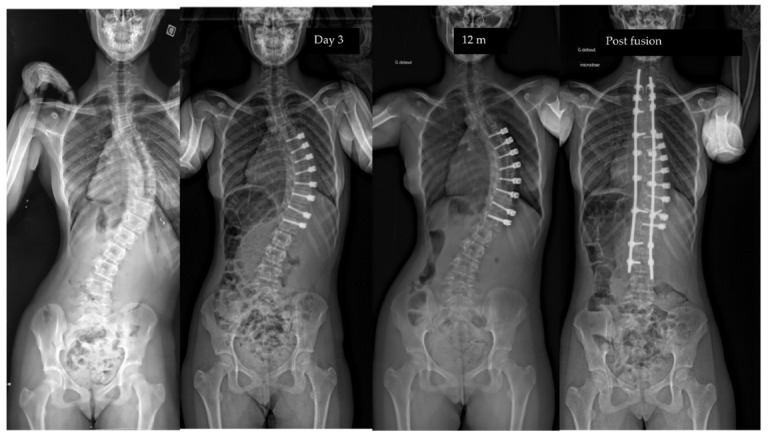
Radiographs of a 12-year-old patient, Risser 0, with a main long thoracic curve (lower end vertebra at L3. The curve measured 53° pre-operatively and was not improved after surgery. An adding-on was observed at 1 y po. A fusion was performed. In this case, 2 mistakes during this index surgery need to be exposed: wrong LIV level (L2 or L3), not enough pre-op correction.

**Table 1 children-10-00192-t001:** Summary of patients’ characteristics. Continuous data are presented as mean (standard deviation).

Patients’ Characteristics				
			*n*	Total
**Age surgery (years)**	12.5 (9; 14)			
**Follow-up (years)**	2.6 (2; 7)			
**Risser**	0	TRC Open	15 (17%)	85
		TRC closed	46 (55%)	
	1		13 (15%)	
	2		11 (13%)	
**Gender**	F		73 (85%)	
	M		12 (15%)	
**Lenke types**	1A		40 (47%)	
	1B		12 (14%)	
	1C		17 (20%)	
	5C		5 (6%)	
	2A		11 (13%)	

**Table 2 children-10-00192-t002:** Comparison of the coronal and sagittal radiological parameters between post-op day 3 and last follow-up (95% of the patients were Risser 3 or more at last f/u; 5% were still at Risser 2 with ongoing significant growth). The curvatures were measured using the Cobb method. The results are expressed as mean and standard deviations.

	Pre-op	Post-op Day 3	Post-op 1y	Last f/u	*p*-Value
**Major curve**	49°(8,9°)	27°(12°)	22°(12°)	19°(14°)	<0.01
**Secondary curve**	27°(14°)	23°(15°)	22°(11°)	17°(10°)	<0.01
**Instrumented curve**	NA	26°(11°)	23°(10°)	20°(14°)	<0.01
**Kyphosis (T1–T12)**	20°(13°)	23°(13°)	25°(12°)	24°(14°)	0.06
**Lordosis (L1–L5)**	36°(12°)	32°(13°)	40°(7°)	40°(8°)	0.07

**Table 3 children-10-00192-t003:** Summary of the complication and revision rates. (thor.: thoracic curve, po: post-operative).

Complications	Patients	Curve Type	Treatment	Time to Diagnosis
**Right shoulder pain**	14 (15%)	10 right thor./5 Double	Painkillers	Immediate post-op
**Aseptic pleural effusion**	1 (1%)	Right thor.	Drainage 2 weeks	45 days po
**Pneumothorax**	2 (2%)	Right thor.	Drainage 2 days	Immediate post-op
**Overcorrection**	10 (11%)	10 right thor.	5 tether release	Between 1,5, and 2 y po
**Tether breakage**	2 (2%)	Lumbar	-	2 y po
**Cranial screw slippage**	6 (7%)	Thor.	-	18 months po
**Curve progression**	5 (2 adding-on) (5%)	Thor.	Fusion	1 y po

**Table 4 children-10-00192-t004:** Comparison of the main published series on VBT in AIS. (OC: overcorrection; PTX: pneumothorax).

Authors	Patients	f/u (y)	Age	Pre-op (°)	Last f/u (°)	Kyphosis Pre-op (°)	Kyphosis Last f/u (°)	Lordosis Pre-op (°)	Lordosis Last f/u (°)	Complications	Revisions
**Samdani 2014** [5]	11	2	12.3	44.2	13.5	20.8	21.6	47.5	54.9	1 atélectasia	2 OC
**Wong 2019** [21]	5	4	11	41.1	32.1	-	-	-	-	1 pneumonia, 2PTX, 2 pleural effusion	1 OC, 1 Fusion
**Alanay 2020** [38]	31	2.2	12.1	46	12	-	-	-	-	2 atelectasia, 1 chylothorax, 1 pleural effusion	2 OC
**Hoernschemeyer 2020** [20]	29	3.1	12.7	49	19	-	-	-	-	1 PTX, 1 syncope,	2 fusions, 4 OC
**Newton 2018** [3]	17	2.5	11	52	27	25	22	-	-	2 atelectasia	4 OC, 4 fusions, 1 rupture
**Newton 2020** [19]	23	3.4	12	53	33	25	19	-	-	1 atelectasia, 1 Horner	3 OC, 3 tether rupture, 1 fusion
**Pehlivanoglu 2020** [18]	21	2	11.1	48.2	10	26.8	26	51.3	51.8	1 chylothorax	1 re-VBT
**Baroncini 2021** [37]	86	2	13.2	52	28.5	28.3	33	47.5	48.4	5 pleural effusion	5 re-VBT/1 irritation psoas
**Samdani 2021** [7]	57	4.6	12.4	40.4	18.7	15.5	19.6	-	-	-	5 OC/2 fusions
**Rushton 2021** [8]	112	3.1	12.7	50.8	25.7		-	-	-	25 complications	15 (7 fusions)
**Courvoisier 2022**	85	2.6	12.5	49	19	20	24	36	40	1 pleural effusion, 2 PTX	5 fusions, 10 OC

## Data Availability

Not applicable.

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
