# Peer review of "Vertebral Body Tethering in AIS Management—A Preliminary Report"

_children, 2023, doi:10.3390/children10020192_

Round 1

Reviewer 1 Report

This study described clinical results of vertebral body tethering in adolescent idiopathic scoliosis, which have been commonly treated with rigid spinal instrumentation if surgical treatment would be required. This paper had abundant cases to discuss the clinical results. There are, however, several points that should be modified before the decision of acceptance.

Line 74; 16 yo is not thought to have enough growth potential. Just as line 79-81, some explanation that the authors deemed this age had enough growth potential would be thoughtful.

Line 161; I think something is missing in the beginning of the sentence.

For Figure 1 to 7, Subheadings and time-points (e.g., 1y post operation) should be described on every radiograph and in the figure legends, respectively.

For Table 2, I was unable to understand the numerals in each parenthesis. Please explain the meaning of the numerals in parentheses.

Author Response

Authors’ response to Reviewer 1 comments

This study described clinical results of vertebral body tethering in adolescent idiopathic scoliosis, which have been commonly treated with rigid spinal instrumentation if surgical treatment would be required. This paper had abundant cases to discuss the clinical results. There are, however, several points that should be modified before the decision of acceptance. 

Line 74; 16 yo is not thought to have enough growth potential. Just as line 79-81, some explanation that the authors deemed this age had enough growth potential would be thoughtful. 

Yes, I have removed the Risser 3 patients from the study to be more consistent with the “growth modulation” philosophy.

Line 161; I think something is missing in the beginning of the sentence. 

Two were excluded because they lacked the minimum required follow-up, so that data from 85 patients were available for this analysis (Figure 3).

For Figure 1 to 7, Subheadings and time-points (e.g., 1y post operation) should be described on every radiograph and in the figure legends, respectively. 

Done

For Table 2, I was unable to understand the numerals in each parenthesis. Please explain the meaning of the numerals in parentheses.

I added the meaning in the legend. “Comparison of the coronal and sagittal radiological parameters between postop day 3 and at last follow-up. The results are expressed as mean and standard deviations. »

Reviewer 2 Report

Thank you for submitting your manuscript on the preliminary results of vertebral body tethering (VBT) in patients with adolescent idiopathic scoliosis (AIS). It was timely and the results interesting. Unfortunately, I have significant concerns regarding publication. My major concerns include:

(1). Although your title states the "French Experience" the combined results are only from two French pediatric orthopaedic programs. However, there were 9 authors from different institutions, including one from Germany. The 92 AIS study patients were described as being from only two centers (Table 3.1). Thus, your study does not represent the French experience, only that of two combined programs.

(2). I am concerned that you included some patients who were Risser 3 because the parents were refractory to fusion. These few patients should have been eliminated as they were probably too mature to benefit from this procedure.

(3). Your results are quite good but mirror what is already known from the recent pediatric orthopaedic spine literature. Also, there was no information regarding how many patients had reached skeletally mature or if you were planning to remove the implants in the future? Thus, there is no truly new clinical relevance at this time. This may change in the future with more follow-up and if the tethers fatigue and break or if the implants are removed. Two years of follow-up is too short to accurately determine initial results for VBT. Skeletal maturity or implant failure should be the first major follow-up point.

Author Response

Authors’ response to Reviewer 2 comments

Thank you for submitting your manuscript on the preliminary results of vertebral body tethering (VBT) in patients with adolescent idiopathic scoliosis (AIS). It was timely and the results interesting. Unfortunately, I have significant concerns regarding publication. My major concerns include:

(1). Although your title states the "French Experience" the combined results are only from two French pediatric orthopaedic programs. However, there were 9 authors from different institutions, including one from Germany. The 92 AIS study patients were described as being from only two centers (Table 3.1). Thus, your study does not represent the French experience, only that of two combined programs.

I agree that it is unclear in the paper. Actually, authors Yan Lefevre5, Federico Solla6, Richard Gouron7, Jean-Damien Métaizeau8 all started their experience 2 years ago in the OR with either me (Center 1) or Vincent Cunin (Center 2). I included their patients in the paper as if they were pertaining to either center 1 or 2. I thought it was fair to include the surgeons we trained in the paper as they reviewed the manuscript and had at least one or 2 patients included in the study. Alice Baroncini (German based fellow) helped in the writing and English editing.

So, I changed the title to: Vertebral Body Tethering in AIS management – A French series, which is maybe less presumptuous.

I also revised the Flowchart to include the patients from the other centers.

(2). I am concerned that you included some patients who were Risser 3 because the parents were refractory to fusion. These few patients should have been eliminated as they were probably too mature to benefit from this procedure.

I agree, I have eliminated these patients from the study.

(3). Your results are quite good but mirror what is already known from the recent pediatric orthopaedic spine literature. Also, there was no information regarding how many patients had reached skeletally mature or if you were planning to remove the implants in the future? Thus, there is no truly new clinical relevance at this time. This may change in the future with more follow-up and if the tethers fatigue and break or if the implants are removed. Two years of follow-up is too short to accurately determine initial results for VBT. Skeletal maturity or implant failure should be the first major follow-up point.

I agree. Since most of our patient have reached Risser 3 or more, I have included this paragraph.

“At last follow-up patients included at Risser 0 TRC closed or above (82%), had all reached Risser 3 or more. 11 (13%) patients included at Risser 0 TRC open have also reached Risser 3 or more at last follow-up. 4 (5%) patients with severe but flexible curves included at Risser 0 TRC open are still at Risser 2 with ongoing significant growth.”  

It is not planned to remove the tether in the future on any of the patient. I added a sentence in M&M.

In our experience, if growth modulation occurs, the risk of curve deterioration is very low. For the oldest patients in our series once skeletal maturity is achieved there the curve is very stable. I agree that longer f/u is required.  

Round 2

Reviewer 2 Report

Thank you for revising your manuscript on vertebral body tethering (VBT). It is improved but I still have concerns regarding publication. These include:

Title

(1). Page 1, Line 2. Your title is improved but still inaccurate. I would suggest adding "A Preliminary Report" since many of your patients are not skeletally mature and still growing, albeit slowly. This indicates a persistent risk for implant breakage, adding on, unplanned return to the operating room (UPROR), and other complications that can affect your results and conclusions. This is true despite institution, country or nation of origin.

Abstract

(2). Page 1, Line 23. Cobb is not an angle, rather a technique used to measure spinal angles (scoliosis, kyphosis, lordosis, etc). Cobb angle is a common term but is jargon. The correct term for scoliosis is "major coronal curve" or just "major curve". Please choose one and use it throughout the text, including any tables, figures or figure legends. See SRS Terminology for further clarification, if necessary.

Materials and Methods

(3). Page 2, Line 83. The term "pure" is inappropriate. Just state "Right Thoracic Curves".

(4). Page 3, Line 118. The term "x-ray" is also a common term that is jargon. Please use "radiograph" or other similar term. This change needs to be made throughout your manuscript and any tables, figures or figure legends where it appears.

(5). Page 3, Line 119. Please add the Risser sign to the "3 y po". Knowing the maturity will be helpful to our readers in understanding your results.

(6). Page 4, Line 122. Please add the lateral view measurements to your legend for Figure 2. This makes the figure easier to appreciate.

Results

(7). Page 5, Line 162. What did you mean by "two"? Were these patients or institutions? Please clarify.

(8). Page 7, Table 2. Please change the term "Cobb" to "major coronal or major".

(9). Page 7, Line 199. Please state your patient's Risser sign at last follow-up. This will demonstrate their maturity (immaturity). It will also indicate that your results are not relatively final. This is a major weakness of your study.

(10). Page 7, Table 3 and Table 4. Consider changing "months" to "years".

Discussion

(11). Page 8, Line 212. Your discussion needs to focus on the similarity between your study and other publications on VBT. Also, are there any significant differences, especially advantageous?

Author Response

Authors responses to reviewer’s comments

Title

(1). Page 1, Line 2. Your title is improved but still inaccurate. I would suggest adding "A Preliminary Report" since many of your patients are not skeletally mature and still growing, albeit slowly. This indicates a persistent risk for implant breakage, adding on, unplanned return to the operating room (UPROR), and other complications that can affect your results and conclusions. This is true despite institution, country or nation of origin.

Ok, I changed the title

Abstract

(2). Page 1, Line 23. Cobb is not an angle, rather a technique used to measure spinal angles (scoliosis, kyphosis, lordosis, etc). Cobb angle is a common term but is jargon. The correct term for scoliosis is "major coronal curve" or just "major curve". Please choose one and use it throughout the text, including any tables, figures or figure legends. See SRS Terminology for further clarification, if necessary.

I used “major curve”. Curvatures were measured “using the Cobb Method” in M&M

Materials and Methods

(3). Page 2, Line 83. The term "pure" is inappropriate. Just state "Right Thoracic Curves".

Done

(4). Page 3, Line 118. The term "x-ray" is also a common term that is jargon. Please use "radiograph" or other similar term. This change needs to be made throughout your manuscript and any tables, figures or figure legends where it appears.

Done

(5). Page 3, Line 119. Please add the Risser sign to the "3 y po". Knowing the maturity will be helpful to our readers in understanding your results.

Done

(6). Page 4, Line 122. Please add the lateral view measurements to your legend for Figure 2. This makes the figure easier to appreciate.

Done (also in figure 7)

Results

(7). Page 5, Line 162. What did you mean by "two"? Were these patients or institutions? Please clarify.

Two “patients”

(8). Page 7, Table 2. Please change the term "Cobb" to "major coronal or major".

Done to “major”

(9). Page 7, Line 199. Please state your patient's Risser sign at last follow-up. This will demonstrate their maturity (immaturity). It will also indicate that your results are not relatively final. This is a major weakness of your study.

Done

(10). Page 7, Table 3 and Table 4. Consider changing "months" to "years".

Done

Discussion

(11). Page 8, Line 212. Your discussion needs to focus on the similarity between your study and other publications on VBT. Also, are there any significant differences, especially advantageous?

The current cohort is one of the largest in the published literature with a minimum 2-year follow-up. Our findings support most of the reported outcomes in the largest published series particularly when comparing the major curve measurements at last follow-up (Table 4): VBT allows a progressive improvement of major curves measurements from 50° to 20°, in patient included around 12-years. Other series had smaller major curve improvement either related to a higher age at surgery[37] or to a lack of initial per-operative correction[3,19,21].

Other differences or advantages (complications for ex.) have been discussed in the rest of the discussion.